# Prevention and Control of COVID-19 in Imperfect Condition: Practical Guidelines for Nursing Homes by Japan Environment and Health Safety Organization (JEHSO)

**DOI:** 10.3390/ijerph181910188

**Published:** 2021-09-28

**Authors:** Sae Ochi, Michio Murakami, Toshihiko Hasegawa, Yoshinori Komagata

**Affiliations:** 1Department of Laboratory Medicine, The Jikei University School of Medicine, Tokyo 105-8461, Japan; 2Division of Scientific Information and Public Policy, Center for Infectious Disease Education and Research, Osaka University, Osaka 565-0871, Japan; michio@cider.osaka-u.ac.jp; 3Research Institute of Future Health, Tokyo 113-0022, Japan; pxn14573@nifty.com; 4Department of Nephrology and Rheumatology, Kyorin University School of Medicine, Tokyo 181-8611, Japan; komagata@ks.kyorin-u.ac.jp

**Keywords:** COVID-19, nursing homes, health management, disaster risk reduction for health, deep defense, infection control

## Abstract

Infection control at nursing homes is a top priority to address the COVID-19 pandemic because people who are the most vulnerable to the pathogen live in close contact. Currently, control measures specifically for nursing homes often ignore under-resourced condition of the facilities. To make guidelines assuming realistic conditions, an expert meeting with 16 members established the key challenges in nursing homes, the basics of infection control, and the major transmission routes. A list of existing guidance was compiled and each item in the list was peer-reviewed by eight experts considering three aspects: significance, scientific validity, and feasibility. Factors related to the nursing home environment, the nature of SARS-CoV-2 transmission, and patient characteristics were identified as the causes of difficulties in infection control at nursing homes. To develop realistic prevention measures in under-resourced condition such as nursing homes, we may need to accept there are no perfect control measures that can achieve zero risk. Instead, the guidelines are based on the concept of deep defense, and practical checklists with 75 items were established. The evaluation of nursing homes by independent organizations using the checklists would be helpful to achieve sustainable infection control.

## 1. Introduction

Healthcare services must be able to continue functioning after a huge disaster. The Sendai Framework for Disaster Risk Reduction endorsed by the United Nations in 2015 includes a global target to ‘reduce disaster damage to critical infrastructure and basic services, among them health and education facilities...’ [1]. This viewpoint is especially important in CBRNE (chemical, biological, radiological, nuclear, and explosive) disasters, during which the fear of invisible pathogens can impact the functioning of health facilities. For example, in Japan, after the Fukushima Daiichi nuclear power plant accident, problems around logistics and human resources in healthcare facilities resulted in the most severe damage, while no literature has described direct impacts of radiation exposure or radiation screening on the functioning of healthcare facilities [2].

Severe functional damage to healthcare facilities from fear of invisible hazards was also observed in pandemic disasters. During the 2009 influenza pandemic in Hong Kong, 50% of hospital staff left the workplace, significantly reducing the productivity of hospitals [3]. After the Ebola outbreak in West Africa in 2013, fear of infection and mistrust in public hospitals caused a severe decrease in the utilization of healthcare services. Pediatric care, antenatal and postnatal care, family planning services, HIV/AIDS and malaria services, general hospital admissions, and major surgeries were particularly affected. As a result, non-Ebola morbidity and mortality rates increased for >1 year [4].

The same is true in the COVID-19 pandemic, an infection caused by the virus SARS-CoV-2. Poor access to care and treatment delay during the pandemic is reported in a variety of services such as dialysis patients [5], malignant tumors [6,7], ophthalmological care [8], etc. To avoid such treatment delay, intervention in healthcare system as a whole is important.

Among these services, nursing homes, where people with high risks live in crowded conditions and perform activities with high infection-exposure risk, are one of the highest priority for intervention [9] because a characteristic of SARS-CoV-2 is that the pathogen disproportionately kills elderly people with comorbidities. Indeed, the rapid spread of infection with high mortality rates (14–33%) among nursing home residents was reported in the early phase of this pandemic and at maximum, and up to a half of deaths from COVID-19 were patients at nursing homes [10,11,12]. In some area the incidence rate ratio for COVID-19 death at long-term care facilities was 13–87 times higher relative to community-living adults over 69 [13].

Currently, this situation has been improving in countries where mass-vaccination is introduced. For example, in United States, number of deaths from COVID-19 decreased from about 6000 cases per day in November 2020 to about 400 cases in September 2021 [14]. Even so, nursing homes remain one of the places where disease clusters occur. In Japan, about a half of the clusters are occurring at medical and welfare institutions [15]. As patients with chronic conditions are more likely to develop severe conditions that consume medical resources, surges in patients at nursing homes can overwhelm local medical facilities. Therefore, establishing effective prevention measures specifically for nursing homes is still an issue of priority [16].

Even so, nursing homes are still one of the most neglected areas of healthcare [17]. There are several existing guidelines for nursing homes issued by governmental bodies [18,19], which describe work restrictions, quarantine, testing, and use of personal protective equipment (PPE) to prevent the spread of infection. However, these guidelines may not take into account the under-resourced situations of nursing homes, and thus not be fully adhered to due to staff shortages [20]. In addition, nursing homes often have few licensed practical nurses and certificated nursing assistants. In such a situation, the guidelines may be too general for staff without the basic medical knowledge to make decisions in practice requiring complex care [21]. For example, they do not specify how frequently the staff and patients should disinfect their hands. Before creating an action plan, we need a simple framework and checklists to implement infection control among patients and staff with heterogeneous backgrounds.

The Japan Environment and Health Safety Organization (JEHSO) [22] was established in September 2020 to support businesses that are struggling in the COVID-19 pandemic. In January 2021, JEHSO established peer-reviewed guidelines and checklists for businesses, including those specific for nursing homes [23], and an associated framework of action. The objective is to establish guidelines for under-resourced institutions with high infection risks. To achieve this objective, this article first identifies specific challenges and major transmission pathways in nursing homes and then summarizes the process of guideline development. Guidelines for the staff for each transmission pathway were also established. Sharing the concepts and contents of the guidelines will contribute to reducing risk and improving security in nursing homes during the COVID-19 pandemic and other pandemics that may occur in the future.

## 2. Materials and Methods

### 2.1. Expert Meetings for Conceptual Framework

Among many guidelines that have been created so far, guidelines with basic concepts and frameworks for action are rare. Therefore, an expert committee was established on March 2020 to establish consensus about the basic concepts related to COVID-19 management. The meeting was organized by an expert on health policy (T.H.). In total, 16 experts with a variety of backgrounds participated in the meeting and made discussion based on the latest issues at the time. Their specialties are listed in Table 1. At each meeting, T.H. summarized the concepts reached by the experts.

The main objectives of the meeting were:To identifying major challenges at nursing homesTo establishing basic concepts of infection controlTo identify priorities of control measures.

### 2.2. Guideline Establishment and Peer Review Process

As a first step, a list summarizing current guidance was made by collecting existing guidelines and checklists from many nations.

Second, each item in the list was peer-reviewed by experts (including the authors) in different fields. The list of the reviewers and their specialties are shown in Table 1. For each item, at least three reviewers independently assessed the draft of guidelines considering three aspects: significance for the control of COVID-19, scientific validity, and feasibility. The aim of the review was to narrow down the items to sustainable, feasible guidelines by eliminating items which had a low evidence level or were difficult to realistically implement. Recommendations that were not feasible (e.g., replacing restaurant condiments after every customer) or not evidence-based (e.g., using cash trays during payment) were excluded according to the results of the peer review.

Third, the results of the assessment were integrated, and then the steering committee finally approved the contents.

## 3. Results

### 3.1. Challenges in Nursing Homes

Challenges in nursing homes were identified based on previous reports and experiences of the experts who participated in the meetings. Three major challenges were identified: challenges related to the nature of nursing homes; challenges related to the nature of SARS-CoV-2; and challenges related to the nature of service users.

#### 3.1.1. Challenges Related to the Nature of Nursing Homes

Infection control in nursing homes is fundamentally different from that in hospitals because of staffing shortages and frequent staff turnover, high users-to-staff ratios, supply shortages, lack of licensed nurses, and less education opportunities for the staff [21]. This means the majority of nursing home staff are not experts in medicine, and few staff members have expert knowledge about infection. Nursing home staff may not be able to read clinical guidelines without investing significant time and effort [24]. Related to this, a significant proportion of staff are part-timers or, in some countries, immigrants, who have few opportunities for training or drills on infection control.

Furthermore, the resources and logistics that are required for effective infection control are often unavailable at nursing homes [24], which makes it difficult to conduct ‘perfect’ infection control.

#### 3.1.2. Challenges Related to the Nature of SARS-CoV-2

Infection control measures in nursing homes usually involve a complete lockdown on individuals suspected of infection, focusing on what can be called “border control”. Another popular measure is frequent polymerase chain reaction (PCR) testing regardless of the existence of symptoms [25,26].

However, such tight regulation may not prevent nosocomial infection because some patients and staff may be infectious even when they have no symptoms and when their PCR tests are negative. There are many cases of nosocomial infection spread by asymptomatic infected individuals [10]. The identification of infected persons is especially difficult among elderly people, who are more likely to be asymptomatic during COVID-19 pneumonia [27].

#### 3.1.3. Challenges Related to the Nature of Service Users

As most of the users of nursing care service are elderly people with morbidity, some characteristics of the users may also make it difficult to implement strict control of infection. For example, many of the service users of nursing homes are elderly with dementia who cannot report their conditions, or those with swallowing disability that often causes aspiration pneumonia. This means slight fever or cough are commonly seen among the users. Thus, identification of COVID-19 symptoms is difficult. Not only that but rejecting users with fever can frequently and considerably reduce their access to services. If nursing homes focus too much on “border control” and continual restrictions on service users, they may reject users with any infectious symptoms, which may weaken mutual trust between the facilities and their users. Anxiety about being rejected may lead users to conceal the onset of fever and other symptoms. This means that strict border control does not provide complete prevention of infection and transmission.

In addition, many users of nursing homes have conditions such as decreased cognitive ability, and it may be difficult to make them comply with rules such as wearing masks, using partitions, and keeping distance.

Lastly, keeping distance between staff members and patients may reduce the quality of nursing service. Therefore, it is also difficult to strictly control transmission spread by patients.

### 3.2. Basic Concepts of Infection Control

Given these challenges, the expert reached the conclusion that infection risk rarely or never becomes zero in nursing homes in current situation. Therefore the risk should be considered as a gradient rather than a threshold.

This situation is similar to the risk of low-dose radiation exposure, which we have experienced in Japan after the Fukushima nuclear accident. Based on our experience, we have learned that ‘how safe is safe’ varies significantly between people. In such condition, showing numbers such as ‘2 m distance’ may not be preferable because people think as if such distance is perfectly safe. It is important to show rough indications, but it seems more important to make people think and judge acceptable risk for themselves [28].

Therefore, we concluded that we need to aim not to eliminate risk, but to make the risk as low as reasonably achievable. To achieve this aim, the following two fundamentals should be shared among nursing home staff: infection risks should be assessed by two axes; and infection risks should be controlled by deep defense under estimation that any control measure cannot be perfect.

#### 3.2.1. Two Axes of Risk Reduction

The risk of severe infection is determined by two axes (Figure 1). One axis describes pathogen exposure level associated with each social activity [29]. The second axis estimates the individual health risks such as older age and complications of chronic diseases. For example, walking outside is associated with lower risk, while nursing and communal living are associated with higher risk. Therefore, infection control should focus on the protection of individuals with higher COVID-19 mortality risk and those who engage in activities that are associated with exposure to high viral load.

#### 3.2.2. Deep Defense under Imperfect Condition

In addition to the basic principles, the guidelines were created with the full realization that each measure individually is incomplete. A high level of protection cannot be attained by implementing a single and strong measure. Therefore, the thinking needs to change from virus eradication to the reduction of virus exposure using multifaceted measures with multiple layers, each of which is imperfect alone.

The concept of deep defense, namely, multiple layers of different and weak protections, is one of the basic principles of safety management of hazards such as radiation [30]. This concept can also be applied to infection management, which requires flexible, timewise, and economically sustainable guidelines. Inspired by the Swiss-cheese model of human error [31], Figure 2 visualizes the concept of deep defense in infection control. Any countermeasures, such as disinfection and keeping distance perfectly, prevent infection transmission. However, depositing multiple layers of such weak protection can provide more effective countermeasures than one strict countermeasure.

### 3.3. Major Transmission Routes and Priorities for Intervention

After establishing these concepts, the expert then identified major transmission routes that cause spread of infection. There are several routes of transmission in nursing homes, where the pathogen is transmitted with high frequency. In these routes, the experts of nursing care identified major transmission routes based on their own experiences. Then other experts discussed and reached consensus about priorities of intervention where prevention measures can be more effectively conducted in under-resourced condition such as nursing homes.

#### 3.3.1. Six “Hot Spots” of Transmission

In the expert meeting, six major transmission routes (“hot spots”) of COVID-19 were identified (Figure 3).

Infection brought in from outside by service users and staff;Transmission between staff members at back roomsOne-to-one transmission between staff and patients;Transmission between users;Transmission between users via surface of staff; andContact transmission from surface of rooms and belongings.

Each route is associated with different measures with different levels of effectiveness and feasibility. As mentioned earlier, control of routes 1, 3, and 4 has limitations because of the nature of the virus, the types of services performed in nursing homes, and patient factors, respectively.

#### 3.3.2. Three Priorities for Intervention

Considering the characteristics and limited resources of nursing homes, the transmission routes 2, 5, and 6 in Figure 3 were identified as priorities for intervention. Recognizing these priority areas may allow the staff to implement measures more efficiently without fatigue.

Prevention of infection transmission between staff members

Even in workplaces where thorough infection control is implemented, the staff tend to relax their attention in staff areas. There have been reports of transmission between employees in break rooms, locker rooms, staff washrooms, or outside of the facilities [32]. The JEHSO guidelines address infection control in staff areas and established the following items.


*“Staff areas, such as employees’ break rooms, are frequently disinfected”*



*“Employees are to avoid eating meals face-to-face”*


2.Prevention of infection transmission between service users via surfaces of staff

Transmission via staff is a type of facility-acquired infection that results in the virus spreading to many patients and should be addressed with high priority. It is necessary to implement thorough preventive measures to restrict infection transmitted by surfaces of the staff, such as hands, gloves, and aprons.

Each preventive measure is common sense, e.g., changing gloves and aprons, disinfecting or washing hands after each patient, and keeping separate flows of clean and dirty items. However, these measures may not be sustainable because the staff at nursing homes have many competing priorities [23]. For example, busy staff may unintentionally take actions such as walking the hallways with an apron worn while performing a task. In another case, busy staff may store clean and dirty items in the cart that holds towels used in body cleaning. Therefore, it is important to maintain an environment which enables all staff members, including part-time employees, to comply with the measures without strain. Our guideline developed specific examples for several items so that it is useful in on-site nursing practice.

3.Environmental disinfection and hand washing

Contact transmission from surface of rooms and belongings is a cause of infection when the virus concentration in the saliva is high. Therefore, environmental disinfection is an important intervention to reduce viral exposure [33]. In contrast to norovirus, a special environmental decontamination or strong disinfectant is not needed to inactivate SARS-CoV-2 [34]. However, this fact is not adequately understood among the public and adverse health effects due to the use of extreme disinfection have been reported [35]. Sufficient inactivation of SARS-CoV-2 has been reported by wiping with commercially-available household products containing both surfactant and up to 70% alcohol. Therefore, it is recommended to select an inexpensive and mild disinfectant that is safe for humans and avoids the dispersion of chemical substances such as ozone in the air.

In addition, the most effective way to prevent contact transmission is to wash or disinfect hands before touching the mouth. Even when a person’s hands are contaminated, contact transmission may not occur unless s/he touches the mouth with the contaminated hands.

### 3.4. Guidelines and Checklists for Nursing Homes

Based on the aforementioned concepts of infection control, simple guidelines for nursing homes that comprise 10 basic concepts were established (Table 2). These principles are important to make all the people on the same table. In this sense, this list is a tool for risk communication as well as scientific guidelines.

In addition, a more detailed checklist with 75 items was made so that independent organizations can evaluate each facility’s compliance to the guidelines (Appendix A). The checklist is further categorized into two parts: items common in service businesses such as restaurants, schools, libraries, etc.; and items specifically for nursing homes. The difference between general and nursing home-specific items are based on the following assumption discussed above.

In nursing homes, service users are often cannot wear masks by themselves due to dementia or other comorbidities.In nursing homes, service users are often cannot report their health condition, such as sore throat, fever, and cough. Therefore, it is difficult to perfectly identify users with COVID-19 symptoms.

The checklist was designed so that evaluators can score the compliance to each item.

## 4. Discussion

Although there already exist a lot of guidelines for healthcare facilities all over the world, the concepts and guidelines in this article are characteristic in that they are based on the premise that risk of infection cannot be eliminated especially in under-resourced conditions such as in nursing homes. COVID-19 is changing from a rapidly expanding pandemic phase to an endemic phase, in which the virus is constantly present in our societies. In such a phase, infection risk never becomes zero and too strict countermeasures such as lockdowns are not always effective. Our guideline repeatedly emphasizes the imperfection of each item so that people do not receive the incorrect idea that there is a silver bullet of infection prevention.

One of the major challenges to infection control in this endemic phase is sustainability of prevention measures. Since the early phase of the COVID-19 pandemic, experts showed concern that infection and quarantine of nursing home staff has the potential to create overwhelming workforce shortages at nursing homes, which already face notoriously high staff turnover rates and have difficulties attracting staff because of low salaries and a demanding work environment [13]. Overly rigorous infection control procedures, repeated waves of infection, and cluster outbreaks may be deteriorating the situation. Even more, if staff are required to follow strict rules that aim at zero risk, the resulting job stress and feelings of being pushed beyond their training may increase the risk of burnout, which may have a negative impact on the prevention of outbreaks [36,37].

This situation resembles to that after the 2011 Fukushima nuclear accident. A systematic review on the impact of the disaster on healthcare facilities revealed that financial problems and staff burnout are major problems in the medium- to long-term phase of the disaster despite minor physical damage [2]. Although clear protection such as PPE is protective against burnout [37], they need to be designed so that they can be implemented without strain to prevent overwhelming of the staff. This is why we have adopted the concept of deep defense, a compilation of weak but easy measures. These guidelines were also designed so that they can be used for risk communication as well as evaluation.

Our guidelines may look less scientific compared to other existing guidelines, because we were careful to avoid the guidelines becoming too scientific. This was because we learned from the previous nuclear disaster in Fukushima that scientific measurements have little effect on prevention of functional damage of healthcare facilities. After the 2011 nuclear accident, a majority of hospital staff left their workplace and sever staff shortage occurred, which lasted for longer than a year [2,38]. Such unwillingness to report for duty seems common in CBRNE disasters, including pandemics, that provoke fear of invisible hazards [39,40]. This functional damage was not caused by the lack of scientific guideposts. Instead, it was caused by differences in risk perception by people, even those within the legal exposure dose limits derived from regulatory science [41]. Because of this difference in perception of ‘how safe is safe’, setting clear safety levels often provoked criticism by those who consider the limit too high [42]. This means that just showing evidence-based facts cannot always prevent social confusion without close communication with on-site staff [28]. In some cases, just showing numbers and figures may provoke a sense of aversion among on-site people because they are talking about not science but their own life [43]. Therefore, any guidelines for risk management should be designed so that they can be used for risk communication as well as risk prevention.

Another challenge in the chronic phase of a pandemic is to prevent indirect health impacts by the disaster. For example, long-term restriction of social activities may cause indirect health impacts on society. After the 2011 nuclear accident in Fukushima, Japan, indoor evacuation and restriction of activities caused an increased risk of diabetes [44], obesity [45], and immobility among the elderly population [46], which can cause excess mortality [44]. Therefore, instead of imposing strict self-restraint continuously and uniformly on the entire society, we may need to prioritize reducing the number of severe infections and deaths in the most vulnerable populations.

In addition, to prevent healthcare collapse and reduce mortality caused by COVID-19 at the local level, control measures should not be left to each facility. Intervention at the local, regional, and national levels is essential to maintain a high quality of care. Therefore, guidelines should be designed so that independent organizations such as local governments can evaluate the quality of service provided by each facility. Evaluators’ guidelines to assess hospital preparedness for disaster were published by the World Health Organization/Pan-American Health Organization [47], but this guideline includes few items for pandemic control. Our checklist is designed so that evaluators can score each item and average scores of each category will show the strength and weakness of the evaluated facility. Such a checklist enables the evaluation of multiple facilities from multilateral viewpoints and enables public health interventions at the local, regional, and national level and thus will improve infection control. We recommend forming a consortium of nursing homes to assess each facility’s infection control measures by using this checklist.

The concepts and guidelines in this article have many limitations. First, it was established by a limited number of experts and did not include skilled nurse practitioners. Second, as this concept was based on our knowledge and experience among Japanese experts, it may not be acceptable from other countries with different culture. Third, scientific evidence of these concepts is weak, though many case reports support this idea.

Even with these limitations, we believe that the concepts and guidelines that openly accept limitation of science can make a difference in risk management and risk communication in nursing homes.

## 5. Conclusions

Infection control at nursing homes is one of the first priorities in the endemic stage of COVID-19. As border control and keeping distance from each other are difficult in the care of institutionalized patients, control measures should focus on preventing transmission between staff members; transmission via surfaces of the staff; and contact transmission from the environment.

To achieve these preventive measures, the concept of deep defense-multiple layers of barriers that are imperfect alone-is the key to sustainable and effective prevention. Our guidelines provide recommendations and checklists that can evaluate nursing facilities from multiple aspects. Our guidelines can contribute to achieving deep defense among nursing homes, which are currently suffering from a lack of resources and effective guidance.

## Figures and Tables

**Figure 1 ijerph-18-10188-f001:**
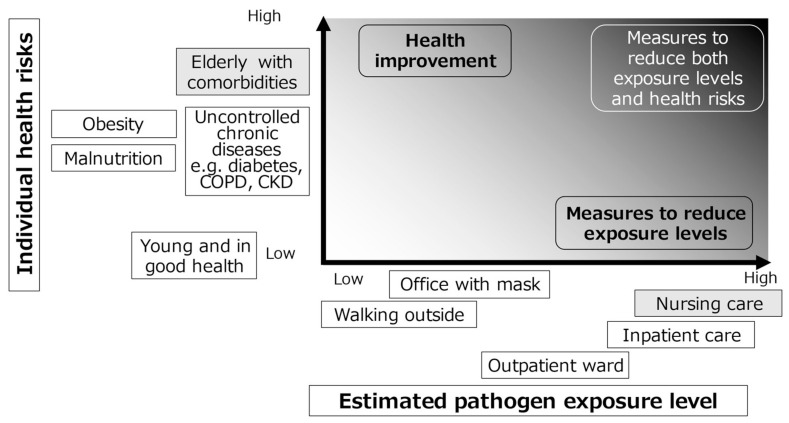
Two determinants of the risk of severe infection and examples of conditions and activities that affects the risks.

**Figure 2 ijerph-18-10188-f002:**
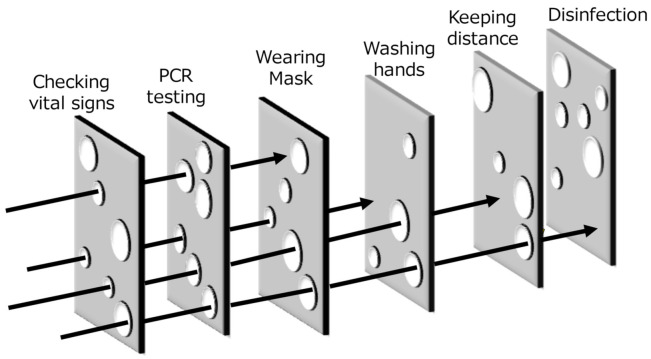
Image of “deep defense” for infection control. Each arrow represents pathogen transmission route. None of the control measures is perfect, but multiple layers of imperfect protections can prevent infection transmission more effectively than one strict countermeasure.

**Figure 3 ijerph-18-10188-f003:**
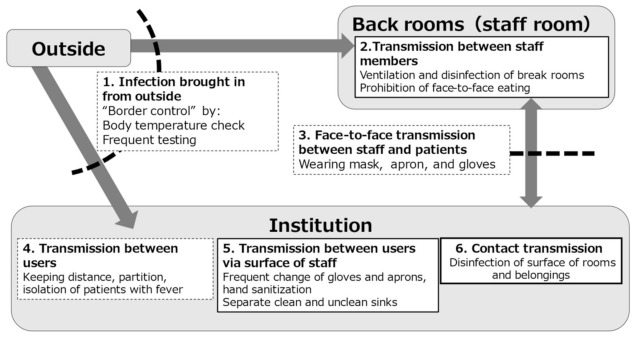
Six major routes of SARS-CoV-2 transmission and possible interventions to target each route. Arrows represent the direction of transmission, and dashed lines represent reduction of the route by the given measures.

**Table 1 ijerph-18-10188-t001:** List of the participants of the expert meeting and the review of the guidelines.

Initial	Specialty	Affiliation, Job
Members of expert meeting		
TH1 *	Health policy, health governance	Former staff of the National Government
SO *	Public health, crisis management	University hospital
GT	Public health, health policy	National Government
SM	Human development	Independent administrative agency
TN	Public health, medical sociology	University hospital
TT	Public health	Publisher
SK1	Medical sociology, nursing care	University hospital
MS	Medical practice	Entrepreneur
TH2	Public health, health economy	University hospital
KH	Public awareness raising	Non-profitable organization
MM1	Anthropology	University
YT	Informatics	University
TT	Business economics	Consultant of healthcare
HY	Nursing care, telemedicine	University hospital
CI	International policy	University
SK2	Journalism	Publisher
Reviewers of the guidelines		
MM2 *	Risk science	University
SO	Disaster public health	University hospital
JK	Ethics	University
AH	In-home medical care	Clinic
Steering committee members		
YK *	Rheumatology	University hospital
MY	Infection control	University
HK	Infection control	University hospital
MK	Infection control	University hospital

* Authors.

**Table 2 ijerph-18-10188-t002:** Ten Principles of SARS-CoV-2 prevention.

Basic Rules of Prevention
1. Put as much distance as far as possible between you and other people as there is no ‘perfectly safe’ distance (though rough indication is 2 m, if not possible, at least 1 m.)
2. Wear a face covering ensuring that it covers the mouth and nose.
3. Always wash or disinfect your hands before touching your face.
4. Talk as quietly as possible to reduce chance of transmission via droplets
5. Clean with a soap or disinfect with bleach or alcohol.
6. Ventilate rooms as often as possible.
7. Do a health check (body temperature check and check whether you have sore throat or cough) immediately before commencing duty, not just in the morning.
Essential Mindset
8. Create an environment in which the infected people are not blamed.
9. Understand that infection control is not perfect but still effective in reducing infection risks.
10. Prepare yourself for the situation that people around you are infected.

## Data Availability

Not applicable.

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
