# Peer review of "Prevention and Control of COVID-19 in Imperfect Condition: Practical Guidelines for Nursing Homes by Japan Environment and Health Safety Organization (JEHSO)"

_ijerph, 2021, doi:10.3390/ijerph181910188_

Round 1
Reviewer 1 Report
Comments
Title: Guidelines for prevention and control of COVID-19 in Nursing Homes by Japan Environment and Health Safety Organization (JEHSO).
This study aimed to share the concepts and contents of the guidelines for nursing homes established by the Japan Environment and Health Safety Organization (JEHSO). The study methodology relied on using a mixed methods approach of primary interviews and secondary data. The presentation of article was generally good with some interesting points made.
However, other areas could do with improvement. Comments are as follows:
In the introduction section, there is little to no reference to similar case studies relating to infection prevention and control in care settings. I think the research gap could be highlighted more effectively by referring to stronger studies. Have a look at the following recent studies as examples:
- Davidson, P. M., & Szanton, S. L. (2020). Nursing homes and COVID‐19: We can and should do better. Journal of clinical nursing.
- Wang, J., Yang, W., Pan, L., Ji, J. S., Shen, J., Zhao, K., ... & Shi, X. (2020). Prevention and control of COVID-19 in nursing homes, orphanages, and prisons. Environmental Pollution, 266, 115161.
Also, in Section 2.1. the authors mention “The meeting was organized by an expert on health policy (T.H.). Participants were experts on public health, global health, disaster medicine, crisis management and nursing care. Meetings were held about once a month from March 2020 to March 2021...” - please provide the exact number of experts that participated in the study and the attendance rate at each meeting based upon the expert input.
I would also like to know whether any nursing home managers/ or nursing home groups were consulted or attended these meetings, as their input into the process would obviously be invaluable in identifying and addressing the issues.
Line 179 – change the word “is” to “are”
In Table 1 of the 10 basic concepts created by the authors as a result of the empirical feedback received through the meetings, how were the 10 concepts ranked? Was it through general consensus or was a scientific methodology applied? Also, the concepts don’t appear to be widely applicable across the nursing care sector. Are there inclusion or exclusion criteria?
It also appears that some of the concepts contradict each other going by their wording. My comments on these concepts are shown below and I welcome the authors response:
- 1.Put as much distance as possible between you and other people(2m, if not possible, at least 1m) - I think the distance should be clearer and based upon the WHO guidelines as they are the leading global body on COVID-19 infection prevention practice. Providing options may cause confusion in terms of implementation. This also contradicts point 2 in Table 2 which simply states “
- Place a mask over your nose.- should this not simply be “wear a face covering ensuring that it covers the mouth and nose”?
- Always wash or disinfect your hands before touching your face.- what about the 5 moments for patient care for healthcare settings? What about after using the toilet?Etc.,?
- Talk as quietly as possible.- how does this relate to COVID-19 infection transmission potential? (I assume perhaps because of the decreased chance for theexpulsion of less droplets?).
- Clean with a soap or disinfect with bleach or alcohol.
- Ventilate rooms as often as possible.
- Do a health check immediately before commencing duty.- what does a health check involve? Lateral flow test? PCR test? Temperature check? Something else?
- Create an environment in which the infected people are not blamed.
- Understand that infection control is effective but not absolutely so.- this sounds very contradictory to earlier principles as my interpretation is that despite our best efforts it won’t be possible to completely prevent COVID-19 transmission? I think this principle weakensthe others.
- Prepare for infection in people around you. - I assume this is related to policies, procedures and training?
Table 2 – the checklist whilst having some scientific validity does not appear to be widely applicable across the care sector and the wording of some of the checklist points needs reviewed. Some of the points could be amalgamated (e.g. 1.4 and 1.5) or scrapped altogether (e.g. 5.2).
There are clear contradictions again regarding the checklist points being made and the general principles of infection prevention and control in care settings. For example, in section 5, point 2 “Employees make efforts, such as asking service users with fever not to enter the facility.” should not be followed as practice. Instead, it should be made clear that anyone displaying COVID symptoms should isolate, get tested and follow the government guidelines as to next steps. This point sounds like it gives the options to service users to enter the facility whether they have a fever or not.
Under the heading “Installation of disinfectants”, “Liquid soap is used for hand washing and dispensers are not refilled by pouring additional soap.” I don’t understand why this would be an issue? As long as people adhere to HH compliance at the correct moments (such as after refilling a soap dispenser, caring around an elderly person or touching surfaces etc.,) then this should not be an issue.
Another contradiction: Under “Mindset of employees; Employees always wear masks when interacting with patients.” this contradicts with “Service users who have a cough are asked to wear masks” and 1.2 “Masks are distributed to users who are not wearing them.” Surely all employees and service users should be wearing a mask regardless as this is just basic infection prevention practice?
Under the point “Employees properly store razors, toothbrushes, and towels for personal use for each patient.” Surely this should be changed to “Employees ensure to prohibit the sharing of personal hygiene items (e.g. razors, toothbrushes, towels) by each patient”?
Discussion – The discussion is not very detailed nor does it refer to previous literature on the topic to support its claims. The weakness of this paper is that the results appear to be based on a small sample of experts in one geographic location. The 10 principles in Table 1 and checklist in Table 2 have some serious flaws in terms of the wording of some of the points.
Author Response
Dear Reviewer1
We sincerely appreciate your constructive comments and advice. We have revised our manuscript following your advice, and please see the following as our point-by-point response.
In the introduction section, there is little to no reference to similar case studies relating to infection prevention and control in care settings. I think the research gap could be highlighted more effectively by referring to stronger studies. Have a look at the following recent studies as examples:
- Davidson, P. M., & Szanton, S. L. (2020). Nursing homes and COVID‐19: We can and should do better. Journal of clinical nursing.
- Wang, J., Yang, W., Pan, L., Ji, J. S., Shen, J., Zhao, K., ... & Shi, X. (2020). Prevention and control of COVID-19 in nursing homes, orphanages, and prisons. Environmental Pollution, 266, 115161.
Response: Thank you for this important advice. We have added some latest articles, including these articles. The followings are the part of our modification.
“There are several existing guidelines for nursing homes issued by governmental bodies [18, 19], which describe work restrictions, quarantine, testing, and use of personal protective equipment (PPE) to prevent the spread of infection. However, these guidelines may not take into account the under-resourced situations of nursing homes, and thus not be fully adhered to due to staff shortage [20]. In addition, nursing homes often have few li-censed practical nurses and certificated nursing assistants. In such situation, the guide-lines may be too general for staff without the basic medical knowledge to make decisions in practice requiring complex care [21].” (Line 72-79)
Also, in Section 2.1. the authors mention “The meeting was organized by an expert on health policy (T.H.). Participants were experts on public health, global health, disaster medicine, crisis management and nursing care. Meetings were held about once a month from March 2020 to March 2021...” - please provide the exact number of experts that participated in the study and the attendance rate at each meeting based upon the expert input.
I would also like to know whether any nursing home managers/ or nursing home groups were consulted or attended these meetings, as their input into the process would obviously be invaluable in identifying and addressing the issues.
Response: Thank you for this question. We have added Table 1 to show the backgrounds of the experts. No managers were officially participated in the meeting, but specialists of nursing care and a consultant of healthcare including nursing homes were involved in the discussion.
Table1. List of the participants of the expert meeting and the review of the guidelines.
|
Initial |
Specialty |
Affiliation, Job |
|
Members of expert meeting |
||
|
TH1* |
Health policy, health governance |
Former staff of the National Government |
|
SO* |
Public health, crisis management |
University hospital |
|
GT |
Public health, health policy |
National Government |
|
SM |
Human development |
Independent administrative agency |
|
TN |
Public health, medical sociology |
University hospital |
|
TT |
Public health |
Publisher |
|
SK1 |
Medical sociology, nursing care |
University hospital |
|
MS |
Medical practice |
Entrepreneur |
|
TH2 |
Public health, health economy |
University hospital |
|
KH |
Public awareness raising |
Non-profitable organization |
|
MM1 |
Anthropology |
University |
|
YT |
Informatics |
University |
|
TT |
Business economics |
Consultant of healthcare |
|
HY |
Nursing care, telemedicine |
University hospital |
|
CI |
International policy |
University |
|
SK2 |
Journalism |
Publisher |
|
Reviewers of the guidelines |
||
|
MM2* |
Risk science |
University |
|
SO |
Disaster public health |
University hospital |
|
JK |
Ethics |
University |
|
AH |
In-home medical care |
Clinic |
|
Steering committee members |
||
|
YK* |
Rheumatology |
University hospital |
|
MY |
Infection control |
University |
|
HK |
Infection control |
University hospital |
|
MK |
Infection control |
University hospital |
Line 179 – change the word “is” to “are”
Response: We have corrected this sentence and other typos in our manuscript.
In Table 1 of the 10 basic concepts created by the authors as a result of the empirical feedback received through the meetings, how were the 10 concepts ranked? Was it through general consensus or was a scientific methodology applied? Also, the concepts don’t appear to be widely applicable across the nursing care sector. Are there inclusion or exclusion criteria?
Response: These 10 items are categorized into two parts: basic rules of prevention (1-6) and essential mindset (7-10). The former is based on scientific evidence, but we reached an agreement that showing ‘how safe is safe’ is not preferable because risks of infection cannot be zero. That is because we often use the words ‘as far as’ and ‘as often as’ instead of using concrete number. We have added the words ‘rough indication’ to make this point clear.
In addition, we explained this point in the main manuscript.
“Given these challenges, the expert reached the conclusion that infection risk rarely or never becomes zero in nursing homes in current situation. Therefore the risk should be considered as a gradient rather than a threshold.
This situation is similar to the risk of low-dose radiation exposure, which we have experienced in Japan after the Fukushima nuclear accident. Based on our experience, we have learned that ‘how safe is safe’ varies significantly between people. In such condition, showing numbers such as ‘2m distance’ may not be preferable because people think as if such distance is perfectly safe. It is important to show rough indication, but it seems more important to make people think and judge acceptable risk for themselves [28].
Therefore, we concluded that we need to aim not to eliminate risk, but to make the risk as low as reasonably achievable. To achieve this aim, the following two fundamentals should be shared among nursing home staff: infection risks should be assessed by two axes; and infection risks should be controlled by deep defense under estimation that any control measure cannot be perfect.” (Line 178-192).
It also appears that some of the concepts contradict each other going by their wording. My comments on these concepts are shown below and I welcome the authors response:
- 1.Put as much distance as possible between you and other people(2m, if not possible, at least 1m) - I think the distance should be clearer and based upon the WHO guidelines as they are the leading global body on COVID-19 infection prevention practice. Providing options may cause confusion in terms of implementation. This also contradicts point 2 in Table 2 which simply states “
Response: As aforementioned, we have added the words ‘rough indication’ to make it clear there is no distance that perfectly assures safety.
- Place a mask over your nose.- should this not simply be “wear a face covering ensuring that it covers the mouth and nose”?
Response: Thank you for this indication. We modified the sentence so that it makes sense in English.
- Always wash or disinfect your hands before touching your face.- what about the 5 moments for patient care for healthcare settings? What about after using the toilet?Etc.,?
Response: This item is for people to be aware that the main reason of washing hands is to prevent hand-to-mouth contact. Even when they wash hands after using the toilet, if they touch a doorknob after the washing and then touch his/her mouth, it may cause infection. We have added this concept in the main manuscript as follows:
“.. the most effective way to prevent contact transmission is to wash or disinfect hands before touching mouth. Even when a person’s hands are contaminated, contact transmission may not occur unless s/he touches the mouth with the contaminated hands.” (Line 285-287)
- Talk as quietly as possible.- how does this relate to COVID-19 infection transmission potential? (I assume perhaps because of the decreased chance for the expulsion of less droplets?).
Response: As you suggested, this is to reduce the chance of droplets transmission. We have added a note in Table 2.
“4. Talk as quietly as possible to reduce chance of transmission via droplets”
- Clean with a soap or disinfect with bleach or alcohol.
- Ventilate rooms as often as possible.
- Do a health check immediately before commencing duty.- what does a health check involve? Lateral flow test? PCR test? Temperature check? Something else?
Response: This meant simple health check by oneself, and we have modified the wording as follows.
“Do a health check (body temperature check and check whether you have sore throat or cough) immediately before commencing duty, not just in the morning.”
- Create an environment in which the infected people are not blamed.
- Understand that infection control is effective but not absolutely so.- this sounds very contradictory to earlier principles as my interpretation is that despite our best efforts it won’t be possible to completely prevent COVID-19 transmission? I think this principle weakens the others.
Response: Thank you for this important advice. This was due to our poor English translation. We tried to say ‘Understand that infection control is not perfect but still effective in reducing infection risks.’ We have modified the words.
- Prepare for infection in people around you. - I assume this is related to policies, procedures and training?
Response: We tried to mean the mindset for all of us, so we have also modified the sentence to ‘Prepare yourself for the situation that people around you are infected.’
Table 2 – the checklist whilst having some scientific validity does not appear to be widely applicable across the care sector and the wording of some of the checklist points needs reviewed. Some of the points could be amalgamated (e.g. 1.4 and 1.5) or scrapped altogether (e.g. 5.2).
Response: Thank you for suggestion. We think 1.4 and 1.5 should be separated, because in Japan, some infection clusters occurred in back rooms instead of bedside. This means even when healthcare staff are very careful at bedside, they sometimes eat together or talk without masks in back rooms. That was why we indicated route 2 (transmission between staff members) in Figure 3 as one of the major transmission routes. To make this point clear, we have added the following sentence in the main manuscript.
“Even in workplaces where thorough infection control is implemented, the staff tend to relax their attention in staff areas. There have been reports of transmission between employees in break rooms, locker rooms, staff washrooms, or outside of the facilities [32].” (Line 253-255)
And we are sorry that item 5.2 omitted some words. We have modified the sentence as follows.
“Employees make efforts as much as possible so that service users with fever not to enter the facility.”
There are clear contradictions again regarding the checklist points being made and the general principles of infection prevention and control in care settings. For example, in section 5, point 2 “Employees make efforts, such as asking service users with fever not to enter the facility.” should not be followed as practice. Instead, it should be made clear that anyone displaying COVID symptoms should isolate, get tested and follow the government guidelines as to next steps. This point sounds like it gives the options to service users to enter the facility whether they have a fever or not.
Response: Thank you for this important pointing-out. We agree that it is ideal that anyone displaying COVID symptoms should be isolated and get tested. However, many of the service users of nursing homes are elderly with dementia who cannot report their conditions or those with swallowing disability that cause chronic coughing, and it is difficult to identify people with COVID symptoms. We have addressed this point in the main manuscript.
“For example, many of the service users of nursing homes are elderly with dementia who cannot report their conditions or those with swallowing disability that often cause aspiration pneumonia. This means slight fever or cough are commonly seen among the users. Thus, identification of COVID-19 symptoms is difficult. Not only that but rejecting users with fever can frequently and considerably reduce their access to services.” (Line 161-165)
Under the heading “Installation of disinfectants”, “Liquid soap is used for hand washing and dispensers are not refilled by pouring additional soap.” I don’t understand why this would be an issue? As long as people adhere to HH compliance at the correct moments (such as after refilling a soap dispenser, caring around an elderly person or touching surfaces etc.,) then this should not be an issue.
Response: As you have pointed out, refilling is not a problem with regard to COVID-19. However, this often cause contamination of other pathogen that can proliferate in soap such as pseudomonas aeruginosa. A steering committee member was anxious that frequent refilling soap for prevention of COVID-19 may increase other healthcare-acquired infection, and we have inserted the sentence. We have added annotation in this part and have added the following sentence at the bottom of the Table.
“* Refilling by pouring additional soap may cause contamination with bacteria that can proliferate in soap, such as pseudomonas aeruginosa.”
Another contradiction: Under “Mindset of employees; Employees always wear masks when interacting with patients.” this contradicts with “Service users who have a cough are asked to wear masks” and 1.2 “Masks are distributed to users who are not wearing them.” Surely all employees and service users should be wearing a mask regardless as this is just basic infection prevention practice?
Response: As we have mentioned above, our assumption is that not all the service users can wear masks, though we agree it is ideal. To address this point, we have modified the title of our manuscript to “Prevention and control of COVID-19 in imperfect condition:…”
Under the point “Employees properly store razors, toothbrushes, and towels for personal use for each patient.” Surely this should be changed to “Employees ensure to prohibit the sharing of personal hygiene items (e.g. razors, toothbrushes, towels) by each patient”?
Response: We appreciate your positive advice. We have followed this and have revised the item.
Discussion – The discussion is not very detailed nor does it refer to previous literature on the topic to support its claims. The weakness of this paper is that the results appear to be based on a small sample of experts in one geographic location. The 10 principles in Table 1 and checklist in Table 2 have some serious flaws in terms of the wording of some of the points.
Response:
Thank you for this constructive advice. We have revised the Discussion to input more information from previous report, and have added the paragraph of limitation as follows.
“The concepts and guidelines in this article have many limitations. First, it was established by a limited number of experts and did not include skilled nurse practitioners. Second, as this concept was based on our knowledge and experience among Japanese experts, it may not be acceptable from other countries with different culture. Third, scientific evidence of these concepts is weak, though many case reports support this idea.
Even with these limitations, we believe that the concepts and guidelines that openly accept limitation of science can make a difference in risk management and risk communication in nursing homes..” (Line 493-501)

Reviewer 2 Report
I believe the subject of the study is interesting as it is an investigation in favor of improving the quality of life and citizen well-being, in this particular case of elderly and frail people at risk of suffering from coronavirus. However, in general, I consider that there is very little depth, especially in the methodology used and the results presented, and conclusions and results are mixed in this section.
Title. I think that the authors have to offer something new to the reader, attractive, that encourages reading, because if the novelty of the "guides for the prevention and control of coronavirus" is not specified, from the beginning, even if it is a sector and location specific, loses interest, since there are numerous guides and manuals published in the same sense.
Abstract. I miss some specific data obtained after the methodology used. I think the concepts here are vague. For example, "a meeting of experts" (line 17) "experts" (line 19). What kind of professionals are they? What is their specialty? How many experts? ...
In line 21, I think it is necessary to clarify that the difficulty of infection control is not only in nursing homes, but is something general or explain why it would be more difficult in these places. On the other hand, it would be necessary to specify or indicate which are the characteristics of the patients that make the process difficult (age, multiple pathological patients, ...).
In line 22, is “to avoid staff burnout” another objective of the study? If so, the text should be structured differently. If not, point it out as a consequence, for example.
On line 24, how many practical checklists were established?
In line 54, I do not consider so strongly that the nursing homes are "obviously the highest priority for intervention". There are other risk groups in society that deserve the aforementioned intervention.
The article lacks concrete data. For example in line 54 I miss a percentage on "mortality rates" that specifies the adjective "high".
Today, in 2021, the situation in nursing homes, depending on the geographical area, has changed significantly as far as COVID-19 is concerned, if we compare it with the previous year. That is why the situation cannot be generalized as an “urgent global issue”.
One of the most interesting aspects of the research is the one indicated in line 76: that guidelines can be established to reduce risk and increase security in nursing homes during pandemics that may occur in the future.
Materials and Methods. On line 79, the number of guidelines is not specified. Guidelines, of what? What are they for? Made by whom? ...
Again, in line 80 is not defined why and by how many experts the “meeting of experts” is formed. It is understandable that the anonymity of the experts is maintained, indicating in some cases their initials, but, at least, their origin, position or job position should be indicated ...
This section could include a summary table that shows the list to which the authors refer on line 89.
In any case, I would try to complete the methodological process with some more tools to give more weight and objectivity to the research.
Results. I don't understand line 113. I think that the adjective “difficult” would come before 'perfect'.
The examples that the authors point out in the text, which are quite clarifying, seem very accurate to me.
Regarding figure 1, I believe that the two axes are well indicated, although it would incorporate more items, both in the “individual health risks” axis and in the “estimated pathogen exposure level” axis.
Figure 3 seems very visual and understandable to me.
In this section, results and consequences appear mixed. It would be necessary to separate the consequences for their corresponding section and leave in this, only, the findings obtained in the investigation.
In the writing, different hypothetical situations are presented as objective (for example in lines 220 and 221) and without specification (lines 225 and 226).
The cost of the measures, is it relevant in the investigation? If so, it should be justified, since it stands out on line 236 (“inexpresive”).
The “common items” seem very general and already known to me.
In my opinion, table 2 would show it as an annex and I would only make a reference to it in this part of the text.
Discussion. The last sentence of this section seems pretentious to me to say that the “checklist enables the universal evaluation”, since each nursing homes, locality, region and nation have their peculiarities.
References. Some of the bibliographic references seem old to me.

Author Response
Dear Reviewer2,
We sincerely appreciate your positive comments as well as advice. We have revised our manusctipt by following your advice. Please see the main manuscript and also the following point-by-point response.
I believe the subject of the study is interesting as it is an investigation in favor of improving the quality of life and citizen well-being, in this particular case of elderly and frail people at risk of suffering from coronavirus. However, in general, I consider that there is very little depth, especially in the methodology used and the results presented, and conclusions and results are mixed in this section.
Title. I think that the authors have to offer something new to the reader, attractive, that encourages reading, because if the novelty of the "guides for the prevention and control of coronavirus" is not specified, from the beginning, even if it is a sector and location specific, loses interest, since there are numerous guides and manuals published in the same sense.
Response: Thank you for this important advice. We agree our main focus is not just the guideline but our concept that assume under-resourced conditions. Therefore, we have modified our title as “Prevention and control of COVID-19 in imperfect condition: practical guidelines for Nursing Homes by Japan Environment and Health Safety Organization (JEHSO)” so that it represents our strength.
Abstract. I miss some specific data obtained after the methodology used. I think the concepts here are vague. For example, "a meeting of experts" (line 17) "experts" (line 19). What kind of professionals are they? What is their specialty? How many experts? ...
Response: We have added the number of the experts (16) in the Abstract and also have added a list of experts, reviewers, and steering committee members as Table 1.
In line 21, I think it is necessary to clarify that the difficulty of infection control is not only in nursing homes, but is something general or explain why it would be more difficult in these places. On the other hand, it would be necessary to specify or indicate which are the characteristics of the patients that make the process difficult (age, multiple pathological patients, ...).
Response: We agree there are many places where intervention was required, such as welfare facilities, prisons, and some festivals. Even so, we think nursing home is still one of the first priorities of intervention to reduce victims. We have added the following sentences to make this point clear.
“Poor access to care and treatment delay during the pandemic is reported in a variety of services such as dialysis patients [5], malignant tumor [6,7], ophthalmological care [8], etc. To avoid such treatment delay, intervention in healthcare system as a whole is important.
Among these services, nursing homes, where people with high risks live in crowded condition and perform activities with high infection-exposure risk, are one of the highest priority for intervention [9] because SARS-CoV-2 is that the pathogen disproportionately kills elderly people with comorbidities. Indeed, the rapid spread of infection with high mortality rates (14-33%) among nursing home residents was reported in the early phase of this pandemic and at maximum, and up to a half of deaths from COVID-19 were patients at nursing homes [10-12]. In some area the incidence rate ratio for COVID-19 death at long-term care facilities was 13-87 times relative to community living adults over 69 [13].” (Line 49-60)
In line 22, is “to avoid staff burnout” another objective of the study? If so, the text should be structured differently. If not, point it out as a consequence, for example.
Response: Thank you for this comment. As you have pointed out, we consider staff’s burnout is a major barrier to sustainability of functions of nursing homes. We have added this point in the manuscript as follows.
“The objective is to establish guidelines for under-resourced institutions with high infection risks.” (Line 87-88)
“Infection control in nursing homes is fundamentally different from that in hospitals because of staffing shortages and frequent staff turnover, high users-to-staff ratios, supply shortages, lack of licensed nurses, and less education opportunities for the staff [21].” (Line 134-6)
“One of the major challenges to infection control in this endemic phase is sustainability of prevention measures. Since the early phase of the COVID-19 pandemic, experts showed concern that infection and quarantine of nursing home staff has the potential to create overwhelming workforce shortages at nursing homes, which already face notoriously high staff turnover rates and have difficulties attracting staff because of low salaries and a demanding work environment [13]. Overly rigorous infection control procedures, repeated waves of infection, and cluster outbreaks may be deteriorating the situation.” (Line 437-442)
On line 24, how many practical checklists were established?
Response: we have added the number (75).
In line 54, I do not consider so strongly that the nursing homes are "obviously the highest priority for intervention". There are other risk groups in society that deserve the aforementioned intervention.
Response: Thank you for this advice. We agree that after implementation of mass-vaccination, it became less obvious. Therefore, we eliminated the sentence and have modified the manuscript as follows.
“Currently, this situation has been improving in countries where mass-vaccination is introduced. For example, in United States, number of deaths from COVID-19 decreased from about 6,000 cases per day in November 2020 to about 400 cases in September 2021[14].” (Line 62-66)
The article lacks concrete data. For example in line 54 I miss a percentage on "mortality rates" that specifies the adjective "high".
Response: Thank you for this advice. We have added the numbers as follows.
“high mortality rates (14-33%) among nursing home residents was reported in the early phase of this pandemic and at maximum, and up to a half of deaths from COVID-19 were patients at nursing homes [10-12]. In some area the incidence rate ratio for COVID-19 death at long-term care facilities was 13-87 times relative to community living adults over 69 [13]. ” (Line 57-61)
Today, in 2021, the situation in nursing homes, depending on the geographical area, has changed significantly as far as COVID-19 is concerned, if we compare it with the previous year. That is why the situation cannot be generalized as an “urgent global issue”.
Response: Thank you for this comment. We agree the situation is rapidly changing, it might no more a global issue, but it still remains an important issue. Therefore, we have modified the manuscript as follows.
“Even so, nursing homes remain one of the places where disease clusters occur. In Japan, about a half of the clusters are occurring at medical and welfare institutions [15]. As patient with chronic conditions are more likely to get severe conditions that consume medical resources, surge in patients at nursing homes can overwhelm local medical facilities. Therefore, establishing effective prevention measures specifically for nursing homes is still an issue of priority [16]. ” (Line66-70)
One of the most interesting aspects of the research is the one indicated in line 76: that guidelines can be established to reduce risk and increase security in nursing homes during pandemics that may occur in the future.
Response: I appreciate this positive comment by the reviewer.
Materials and Methods. On line 79, the number of guidelines is not specified. Guidelines, of what? What are they for? Made by whom? ...
Again, in line 80 is not defined why and by how many experts the “meeting of experts” is formed. It is understandable that the anonymity of the experts is maintained, indicating in some cases their initials, but, at least, their origin, position or job position should be indicated ...
This section could include a summary table that shows the list to which the authors refer on line 89.
Response: As you have mentioned, some experts do not like to make their names and affiliations clear. So we have shown initials, rough specialties, and affiliations of the members as Table 1. In total, 16 experts joined in the meeting, 4 in guideline review, and 4 in approval of the guideline.
In any case, I would try to complete the methodological process with some more tools to give more weight and objectivity to the research.
Response: Thank you for this important advice. We have thoroughly rewritten this part with a view of objectivity. Chapter and categorization were also revised so that they correspond to the chapters of the Result. Please see the revised manuscript.
Results. I don't understand line 113. I think that the adjective “difficult” would come before 'perfect'.
Response: We have modified the sentence as follows.
“, which makes it difficult to conduct ‘perfect’ infection control.”(Line XXX)
The examples that the authors point out in the text, which are quite clarifying, seem very accurate to me.
Response: I appreciate this positive comment by the reviewer.
Regarding figure 1, I believe that the two axes are well indicated, although it would incorporate more items, both in the “individual health risks” axis and in the “estimated pathogen exposure level” axis.
Response: Thank you for this advice. We have added some conditions and activities related to nursing homes. We have also modified the explanation of the figure as follows.
“Figure 1. Two determinants of the risk of severe infection and examples of conditions and activities that affects the risks.”
Figure 3 seems very visual and understandable to me.
Response: I appreciate this positive comment by the reviewer.
In this section, results and consequences appear mixed. It would be necessary to separate the consequences for their corresponding section and leave in this, only, the findings obtained in the investigation.
In the writing, different hypothetical situations are presented as objective (for example in lines 220 and 221) and without specification (lines 225 and 226).
Response: We agree the chapters was not properly separated. We have revised the Result and have tried to make it clear which are the results of discussion and which are contents of the product (guideline).
The cost of the measures, is it relevant in the investigation? If so, it should be justified, since it stands out on line 236 (“inexpensive”).
Response: Thank you for this comment. We agree that our objectives were not obvious. Our aim is to establish guidelines for under-resourced institutions with high infection risks, so cost is important. We have addressed this point in Line XXX, and as aforementioned, we have also modified the title of this article.
The “common items” seem very general and already known to me.
Response: Thank you for this positive comment. Although these items look commonsense for people who know risks better, it is still important to clearly state that risk cannot be zero in any situation and that creating mindset is important. In this sense, these are tools for risk communication as well as guidelines. We have added this point in the main manuscript.
“These principles are important to make all the people on the same table. In this sense, this list is a tool for risk communication as well as scientific guidelines.” (Line XXX)
In my opinion, table 2 would show it as an annex and I would only make a reference to it in this part of the text.
Response: Thank you for this advice. We have moved the former Table2 to supplementary material.
Discussion. The last sentence of this section seems pretentious to me to say that the “checklist enables the universal evaluation”, since each nursing homes, locality, region and nation have their peculiarities.
Response: We are sorry this is a of translation. We intended to say “evaluation from multilateral viewpoints” and have corrected the words.
“Such a checklist enables the evaluation of multiple facilities from multilateral viewpoints..” (Line 489-490)
References. Some of the bibliographic references seem old to me.
Response: Through overall revision of the article, references also have revised and have included more articles.
Reviewer 3 Report
Dear Authors,
thank you for the possibility to review your article.
The topic is interesting. The study is clear and well conducted but, in my opinion, vaccination cannot be omitted among the tools of SARS-CoV2 prevention and control; for this reason, I suggest to insert vaccination as the main preventive measure, before analyzing daily prevention.
Kind regards
Author Response
Dear Reviewer3,
We sincerely appreciate your positive advice. We have modified the manuscript following your advice. Please see the attached manuscript as well as the following response.
The topic is interesting. The study is clear and well conducted but, in my opinion, vaccination cannot be omitted among the tools of SARS-CoV2 prevention and control; for this reason, I suggest to insert vaccination as the main preventive measure, before analyzing daily prevention.
Response: Thank you for this important suggestion. We agree that the situation is rapidly changing and now we cannot ignore importance of vaccination. However, at the time of the establishment of this guideline, implementation of mass-vaccination was not fully foreseeable. In addition, there still exist ethical issue related to vaccination, such as freedom of choice and we thought it difficult to encourage vaccination in a uniform way. Therefore we have omitted this topic from these guidelines. Even so, we have added the following sentences to address the importance of vaccination.
“Currently, this situation has been improving in countries where mass-vaccination is introduced. For example, in United States, number of deaths from COVID-19 decreased from about 6,000 cases per day in November 2020 to about 400 cases in September 2021. Even so, nursing homes remain one of the places where disease clusters occur. In Japan, about a half of the clusters are occurring at medical and welfare institutions.” (Line 62-66)
Reviewer 4 Report
The paper entitled “Guidelines for prevention and control of COVID-19 in Nursing Homes by Japan Environment and Health Safety Organization (JEHSO) ” aimed to share the concepts and contents of the guidelines for nursing homes established by the Japan 16 Environment and Health Safety Organization (JEHSO).
It is interesting and timely work. The conclusions are largely sound and improve the existing knowledge.
On my opinion it don't need revision
Author Response
We appreciate your positive comment.
We have modified the manuscript following other reviewers’ comments, and appreciate if you look over the latest version.
Round 2
Reviewer 3 Report
Dear Authors,
Thank you for the explanation you give me.
My comment was born from your indication on the time under examination (March 2020 - March 2021) therefore also when vaccination had already been introduced.
Kind regards